# The braid index of DNA double crossover polyhedral links

**Xiao-Sheng Cheng[1,2]☯, Yuanan Diao [2]☯ ***

**1** School of Mathematics and Statistics, Huizhou University, Huizhou, Guangdong, P. R. China, **2** Department of Mathematics and Statistics, University of North Carolina at Charlotte, Charlotte, NC, United States of America

☯ These authors contributed equally to this work.
* ydiao@uncc.edu

## Abstract

In this paper, the authors study the mathematical properties of a class of alternating links called polyhedral links which have been used to model DNA polyhedra. The motivation of such studies is to provide guidance and aid in the research of the properties of certain DNA molecules. For example, such studies can provide characterizations of the structural complexity of DNA molecules. In an earlier work, Cheng and Jin studied the mathematical properties of such polyhedral links and were able to determine the braid index of a double crossover polyhedral link with 4 turn. However, the braid index of a double crossover polyhedral link with 4.5 turn remained an unsolved problem to this date, even though the graphs that admit the double crossover polyhedral links with 4.5 turn have been synthesized. In this paper, we provide a complete formulation of the braid index of a double crossover polyhedral link with an arbitrary turn number. Our approach is more general and it allows us to completely determine the braid indices for a much larger class of links. In the case of the double crossover polyhedral links, our formulation of the braid index is a simple formula based on a simpler graph used as a template to build the double crossover polyhedral links.

**Data Availability Statement:** This is a method/ theory only paper, no data was generated nor needed for its production. All relevant information is contained in the manuscript.

**Funding:** Unfunded studies: YD XC: Natural Science Foundation of Guangdong Province, China (No. 2016A030313122) XC: Excellent youth

## 1 Introduction

The synthesis of topologically interesting structures like braids in the range of nanometer to micrometer is becoming popular. For example, braiding of nanofibers in supramolecular gels [1], Molecular braids in metal-organic frameworks [2], and knotted hydrocarbon complexes [3]. The topological properties of these chemical and biological braid structures are of great interests in research. Braid index, a fundamental topological invariant that is sometimes used to describe the complexity of a molecule, is another potential tool that can be used to study the complexity properties of certain DNA molecules, some of which have been synthesized in laboratories by chemists and biologists in recent years. For example, through four arm immobile DNA crossover junctions, the following DNA polyhedral links with polyhedral shapes have been synthesized in laboratories: DNA cube [4], DNA tetrahedron [5], DNA octahedron [6], DNA truncated octahedron [7], DNA bipyramid [8], DNA dodecahedron [9], and DNA dodecahedron and buckyballs [10].

training project of Huizhou University (No. 20160224082617206) XC: Training Program of the Major Research Plan of Huizhou University (No. 20170327011726271).

**Competing interests:** The authors have declared that no competing interests exist.

**Fig 1. [21] Top: DNA polyhedra built with 3-point star motifs (tetrahedron, cube, dodecahedron and buckyball); Bottom: DNA polyhedra built with 4-point and 5-point star motifs (octahedron and icosahedron).**

A common strategy used to build/assemble more complicated DNA polyhedra is to use simpler structures such as a double crossover as building blocks [10–20]. For example, Zhang *et al.* [18] synthesized 4.5 turn cubes in a laboratory by "*n*-point star motif (tiles)". (Fig 1 shows a few regular DNA polyhedra built using 3-point or 4-point star motifs.) In the case of the 4 or 4.5 turn cubes (shown in Figs 2 and 3), each of the eight vertices of the cube is replaced by a three-point-star tile and each face (a square) of the cube consists of four three-point-star tiles. Such conditions cannot be met by adjusting the concentration and/or flexibility of the DNA tiles.

In [21, 22], Cheng *et al.* studied the braid index problem for several polyhedral links that were proposed by mathematicians as potential candidates as DNA polyhedra to be synthesized. For the few relatively simple ones with single crossover they were able to determine the braid indices of these links completely. They were also successful in determining the braid indices of a more complicated one, namely the double crossover polyhedral links with 4 turn [21].

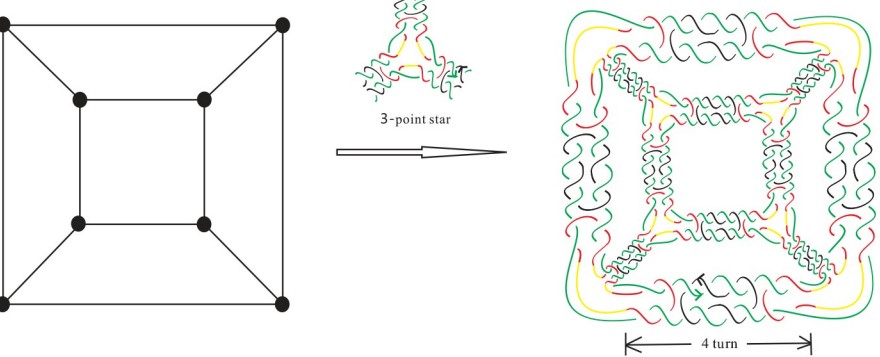

**Fig 2. An alternating link $\mathcal{L}_0$ with negative writhe that is the realization of a 4 turn double crossover cube link.**

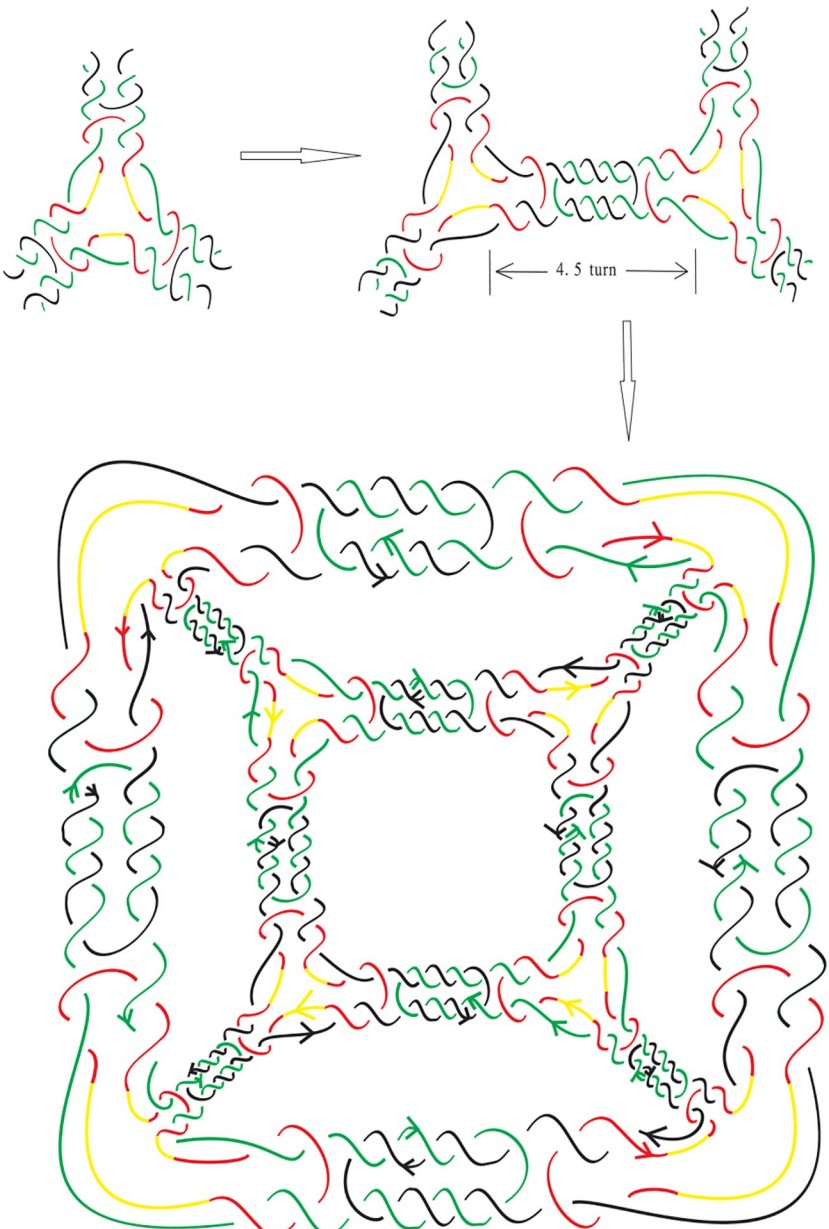

**Fig 3. An alternating link $\mathcal{L}_0^*$ with negative writhe that is the realization of a 4.5 turn double crossover cube link.**

However the braid indices of polyhedral links with 4.5 turn double crossover remain unsolved until now. This is the motivation of this paper. Here, we present a solution to this problem as the consequence of a much more general result, that is, we present a solution that would allow us to determine the braid index of a double crossover polyhedral link with an arbitrary turn number.

Alexander proved that every oriented link can be represented as a closed $n$-string braid in which all strings in the braid are assigned parallel orientation [23]. The braid index $\mathbf{b}(L)$ of a link $L$ is the least number $n$ of strings needed to present $L$ a closed braid. Yamada [24] proved that the braid index of a link $L$ is bounded above by the number of Seifert circles in any given

regular diagram of $L$. Consequently, $\mathbf{b}(L)$ equals the minimum number of Seifert circles among all link diagrams of $L$. However, the braid indices for most links remain unknown due to the lack of a universal method/algorithm that can guarantee the successful determination of the braid index of any link. Thus it makes sense for one to seek general lower and upper estimates of $\mathbf{b}(L)$. One such result is due to Franks and Williams [25], and Morton [26]. Specifically, let $P_L(a, z)$ be the HOMFLY-PT polynomial of a link $L$ and $\gamma(L)$, $\alpha(L)$ be the maximal and minimum powers of $a$ in $P_L(a, z)$ respectively, then

$$\frac{1}{2} \, \text{span}_a P_L(a, z) + 1 \leq \mathbf{b}(L), \tag{1}$$

where $\text{span}_a P_L(a, z) = \gamma(L) - \alpha(L)$. The inequality in (1) is known as the *MFW inequality*.

Of course, in the case that the equality in the MFW inequality holds (which we shall call it the *MFW equality*), one obtains the braid index of the link in question. Much effort was devoted to the identification of links that satisfy the MFW equality. For example, it was shown that the MFW equality holds for torus links and closed positive braids with at least one full twist [25]. In fact Franks and Williams first conjectured that the MFW equality holds for any closed positive braid. This conjecture was found to be false: a counter example was later given by Morton and Short using a 2-cable of the trefoil [27]. More work regarding closed positive braids can be found in [28], where Nakamura identified families of (infinitely many) positive closed braids for which the MFW equality either holds or fails. In [29], Elrifai classified all 3-braids for which the MFW equality holds. Murasugi [30] proved that the MFW equality holds for all rational links and alternating links that are fibered. However the MFW equality does not hold in general for all alternating links due to counter examples discovered by Murasugi and Przytycki [31]. Recently, Diao and colleagues [32, 33] used a diagrammatic approach to establish the MFW equality for a large class of alternating links that includes all alternating pretzel links and Montesinos links, leading to the complete determination of the braid index of any such link. It is important to note that none of these known results provides answers to the polyhedra links discussed in this paper.

The main result of this paper is the determination of the braid index for any link in a large class of positive (or negative) alternating links. This link class includes all double crossover polyhedral links with any given turn number. More specifically, we prove that the MFW equality holds for any link from this class. An immediate consequence of this main result is the solution to the open braid index problem for double crossover polyhedral links with 4.5 turn. In addition to providing the determination of an important measure for characterizing and analyzing the structure and complexity of DNA polyhedra modeled by the double crossover polyhedral links (with any given turn number), our research can also be used as tools in the study of topological entanglement of more general biopolymers encountered in DNA nanotechnology.

The rest of the paper shall be organized in the following way. In the next section, we provide some necessary background knowledge, concepts and terminology in knot theory and graph theory. In Section 3, we outline the results for the double crossover polyhedral links. The reason for us to do so, instead of stating the theorems in the general cases only, is so that our reader familiar and interested in the applications of these types of links can easily comprehend our results and compare them with the previously known results. In Section 4, we state and prove our theorems under in the general cases. In the last section, we end the paper by showing how our approach and results in this paper may be used to provide alternative proofs for some previously known results in the case of double crossover polyhedral links with 4 turn, as well as obtaining some parallel (new) results for the case of double crossover polyhedral links with 4.5 turn.

## 2 Basic background knowledge, concepts and terminology

### 2.1 Knot theory

A *link* consists of several simple closed curves embedded in the 3-space $\mathbf{R}^3$ where each of these closed curves is called a *component* of the link. A link with one component is also called a *knot*. A link is said to be *oriented* if each of its components is assigned an orientation. A *regular diagram*, or just a diagram, of a link is the projection of the link (as a set of disjoint, closed simple space curves) onto a plane in which strands can cross each other only transversely and at most two strands are allowed to cross at the same point. A point at which two strands cross each other is called a *crossing point* (or just a crossing for short). In such a projection, the under-strand and upper-strand at each crossing are specified so that the original link can be re-constructed from the (projection) diagram. The *crossing number* of a link $L$, denoted by $c(L)$, is defined as the least number of crossings in any regular diagram of the link. A diagram with the least number of crossings for a given link is called a *minimal* diagram of the link. A crossing in a link diagram is said to be *nugatory* if the crossing is as shown in Fig 4, which can be removed by a simple twist on a part of the diagram. A link diagram is said to be *alternating* if one encounters the crossings alternately between over strand and under strand when traveling along any component of the link following any orientation. A link is said to be *alternating* if it has a regular projection that is alternating. It is a well known result (as a consequence of the Jones polynomial) that an alternating link diagram without nugatory crossings is a minimal link diagram.

An *n*-string braid $\beta$ is an *n*-string tangle diagram with fixed end-points as shown in Fig 5. The closure of braid $\beta$ as shown in Fig 5 is called a closed braid, and denoted by $\hat{\beta}$. It is known that every oriented link can be represented as a closed braid with the strings in the braid assigned parallel orientation [23]. The *braid index* of an oriented link $L$, denoted by $\mathbf{b}(L)$, is defined as the least number of strings needed to present $L$ as a closed braid. It is obvious that $\mathbf{b}(L) = \mathbf{b}(L^*)$ if $L^*$ is the mirror image of $L$.

The HOMFLY-PT polynomial is an invariant of oriented links, introduced in [34] and [35] independently. Let $L$ be an oriented link and $D$ be a regular projection of $L$. Let $D_+$, $D_-$, and $D_0$ be oriented link diagrams that coincide with each other except at a small neighborhood of a crossing as shown in Fig 6. The HOMFLY-PT polynomial of an oriented link $L$, denoted by $P_L(a, z)$, is a two variable Laurent polynomial with integer coefficients satisfying the following conditions:

$$P_{D_1}(a, z) = P_{D_2}(a, z) = P_L(a, z) \tag{2}$$

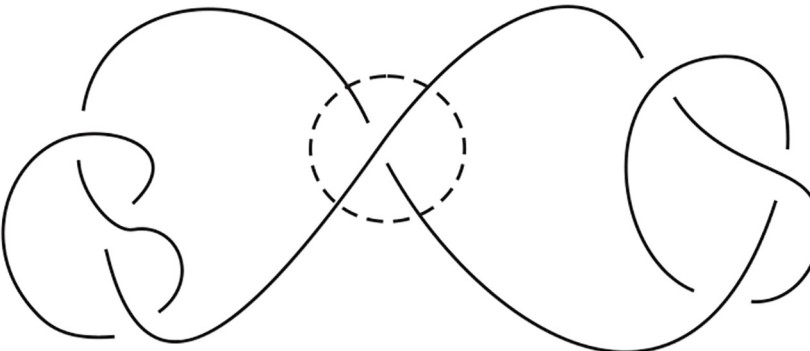

**Fig 4. A nugatory crossing.**

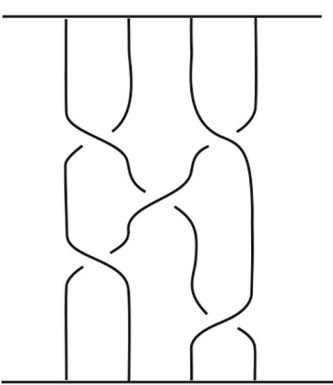
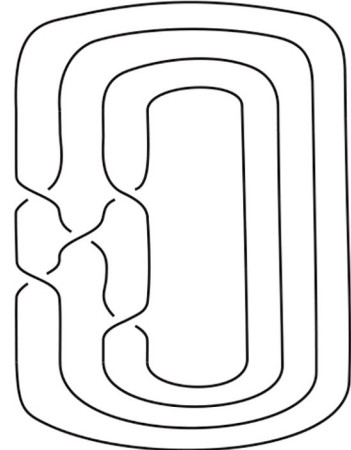

**Fig 5. A braid and its closure.**

if $D_1$ and $D_2$ are both regular projections of the same link $L$;

$$aP_{D_+}(a, z) - a^{-1}P_{D_-}(a, z) = zP_{D_0}(a, z) \qquad (3)$$

and $P_D(a, z) = 1$ if $D$ is a regular projection of the unknot.

Eq (3) in the above is called the *skein relation* of the HOMFLY-PT polynomial, which can be rewritten as the following two equivalent forms:

$$P_{D_+}(a, z) = a^{-2}P_{D_-}(a, z) + a^{-1}zP_{D_0}(a, z), \qquad (4)$$

$$P_{D_-}(a, z) = a^2 P_{D_+}(a, z) - azP_{D_0}(a, z). \qquad (5)$$

It can be easily shown that $P_D(a, z) = P_{D^*}(-a^{-1}, z)$, where $D^*$ is the mirror image of $D$. This implies that

$$\mathrm{span}_a P_L(a, z) = \mathrm{span}_a P_{L^*}(a, z).$$

## 2.2 Notations and terminology in graph theory

We assume that our reader will have some knowledge in graph theory hence this subsection will only provide a list of notations and terminologies used in this paper for the purpose of easy referencing. The required knowledge is basic and elementary, and can be found in any graph theory textbook. Let $G$ be a graph. The following is a list of notations concerning $G$:

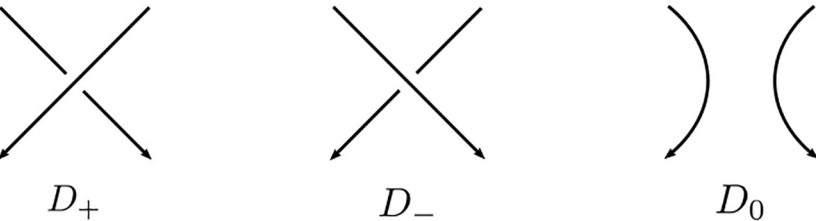

**Fig 6. The sign convention at a crossing.**

- $V(G)$ and $E(G)$: the vertex set and edge set of $G$ respectively;

- $|V(G)|$ and $|E(G)|$: the number of vertices and the number of edges in $G$ respectively;

- $\kappa(G)$: the number of connected components of $G$;

- $G - e$: the graph obtained from $G$ by deleting the edge $e$;

- $G/e$: the graph obtained from $G$ by contracting the edge $e$ (namely deleting $e$ first and then identifying its two end vertices);

- a *bridge* edge $e$: an edge $e$ satisfying the condition $\kappa(G - e) > \kappa(G)$;

- a *loop* edge $e$: an edge $e$ with its two end-vertices being the same;

- degree of a vertex $v$ ($d(v)$): the number of edges connected to $v$;

- a *k-regular* graph: a graph in which every vertex has degree $k$;

- a *bipartite* graph: a graph whose vertices can be partitioned into two non-empty sets such that no edge of $G$ is between vertices belonging to the same set;

- a *simple* graph: a graph in which any pair of vertices can be connected by at most one edge;

- a *path* in a graph: an (ordered) sequence of distinct vertices such that two adjacent vertices is connected by an edge but no other pairs of edges are connected by any edges;

- a *cycle* in a graph: a path in the graph with an additional edge added that connects the first and the last vertices in the path;

- a *planar* graph: a graph that can be embedded in a plane such that edges will not cross each other;

- a *plane* graph: a specific embedding of a planar graph in a plane.

It is well known that a graph is bipartite if and only if every cycle contained in the graph (as a subgraph) has even length.

## 2.3 The Seifert graph of an oriented link diagram

Given a diagram $D$ of an oriented link $L$, if we "smooth" every crossing in $D$ (as in the case of $D_0$ in Fig 6), then we obtain a collection of disjoint topological circles called *Seifert circles*. One can construct a graph $G_D$ in which each vertex corresponds to a Seifert circle of $D$, and two vertices in $G_D$ are connected by $k$ edges if there are $k$ crossings between the two Seifert circles corresponding to these vertices. Fig 7 shows an oriented link diagram, its Seifert circle

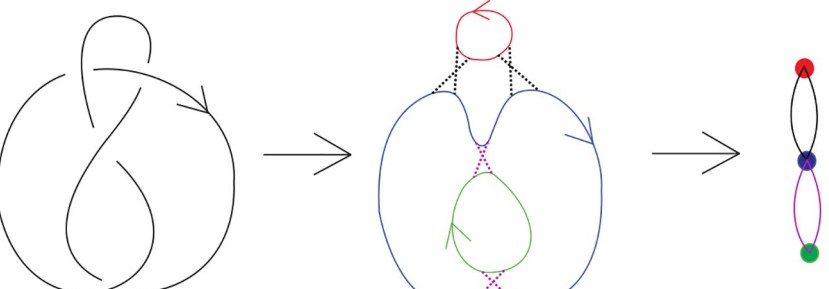

**Fig 7. The figure-eight knot, its Seifert circle decomposition and the corresponding Seifert graph.**

decomposition and its corresponding Seifert graph. It is easy to prove that the Seifert graph of any link diagram is bipartite (hence loopless) and planar. On the other hand, given any plane bipartite graph $G$, one can construct an alternating link diagram $D$ such that its Seifert graph $G_D$ is $G$.

## 3 The main results for double crossover polyhedral links

DNA is a double helix formed by base pairs attached to a sugar-phosphate backbone. The orientations of the two strands in the double helix are antiparallel. A nanostructure used by DNA can have many link component pairs with antiparallel orientations. A double crossover polyhedral link $L$ is an alternating link that is constructed from its Seifert graph by the following procedure. We first start from a simple plane graph $G$ that is loopless, called the *template graph*. We then construct the Seifert graph of the link $L$ from $G$ by replacing its vertices and edges with some particular kind of cycles. More specifically, we define the following types of cycles. A Type (1A) cycle is a cycle of even length with two of its vertices marked (we call these vertices *attaching vertices*), and the two paths between them both have odd length. A Type (1B) cycle is a cycle of even length with two attaching vertices, and the two paths between the attaching cycles both have even length. Finally, a Type (2) cycle of degree $j \geq 2$ has $j$ attaching vertices, and the path between any two adjacent attaching vertices contains an even number of edges. We note that the cycles and the paths between adjacent attaching vertices of these cycles can have different lengths. We can now construct two types of double crossover polyhedral links $L(G)$ by constructing their Seifert graphs $G^*$ first from a template graph $G$ as shown in Fig 8, followed by detailed descriptions.

*Type A double crossover polyhedral link.* Let $G$ be a simple bipartite plane graph and we construct the Seifert graph $G^*$ of $L$ as follows: each edge of $G$ is replaced with a Type (1A)

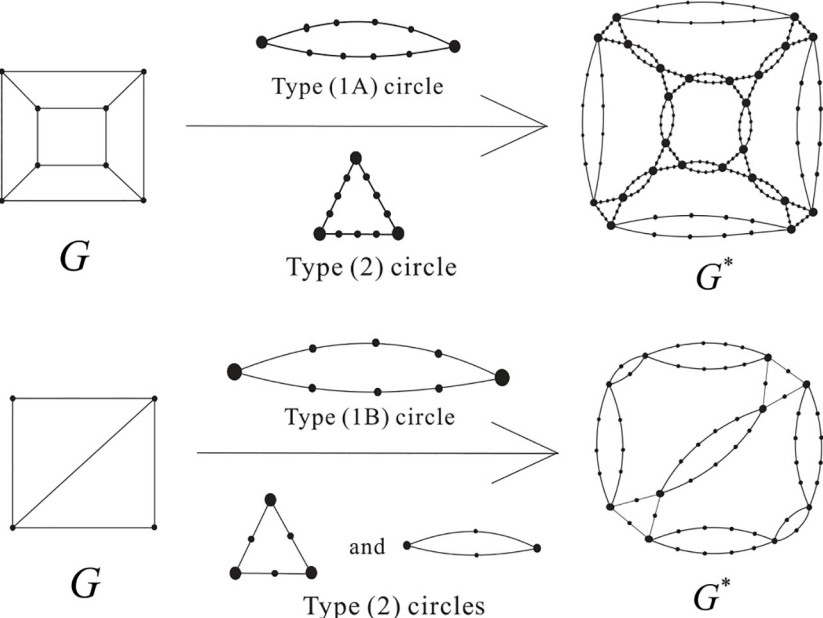

**Fig 8. The attaching vertices are marked by large dots.** Top: how $G^*$ is constructed from the graph $G$ by using 12 Type (1A) circles with length 10 and 8 Type (2) circles with length 12 and 3 attaching vertices. Bottom: how $G^*$ is constructed by using Type (1B) circles of length 8 and Type (2) circles either with length 6 and 3 attaching vertices, or with Type (2) circles with length 4 and 2 attaching vertices.

cycle, and each vertex of degree $j \geq 2$ in $G$ is replaced by a Type (2) cycle with $j$ attaching vertices, and then identifying the corresponding attaching vertex pairs in a natural way.

**Type B double crossover polyhedral link**. Ditto the construction of $G^*$ for a Type A double crossover polyhedral link above, however in this case Type (1B) cycles are used in the place of Type (1A) cycles and $G$ does not have to be bipartite.

Notice that it is a necessary condition for $G$ to be bipartite in the case of a Type A double crossover polyhedral link (since $G^*$ is bipartite). An alternating link $L(G)$ so constructed is apparently minimum as it has no nugatory crossings. A double crossover cube link with 4.5 turn is a Type A double crossover polyhedral link, while a double crossover cube link with 4 turn is a Type B double crossover polyhedral link. The top of Fig 8 shows how the Seifert graph of a double crossover cube link with 4.5 turn is constructed from a 3-regular template graph $G$ using Type (1A) cycles of length 10 and Type (2) cycles of length 12. The Seifert circle decomposition of the corresponding link, which is the link shown in Fig 3, is shown in Fig 9. The bottom of Fig 8 shows how the Seifert graph of a double crossover cube link with 4 turn is constructed from a template graph $G$ that is not bipartite nor regular using Type (1B) cycles of length 8 and Type (2) cycles either with length 6 and 3 attaching vertices, or with Type (2) cycles with length 4 and 2 attaching vertices.

A main motivation of this paper is to solve the braid index problem for a double crossover polyhedral link $L(G)$ with 4.5 turn, we have succeeded in achieving this goal. In fact, we have obtained more general results concerning the braid index of a double crossover polyhedral link of any given turn number. In particular, for some special classes of $G$, we can express $\mathbf{b}(L(G))$ in terms of a simple formula using the numbers of vertices and edges in $G$. We state one such result below.

**Theorem 1** *Let $G$ be a simple, $k$-regular ($k \geq 3$) plane graph. If $G$ is bipartite and $L(G)$ is a Type A double crossover polyhedral link with $G$ as its template graph and by replacing its edges by Type (1A) cycles of length $2m_1$ with $m_1 \geq 2$, and replacing its vertices by Type (2) cycles of length $2km_2$ with $m_2 \geq 1$, then $\mathbf{b}(L(G)) = (km_2 + 1)n(G) + (m_1 - 1)e(G)$. On the other hand, if*

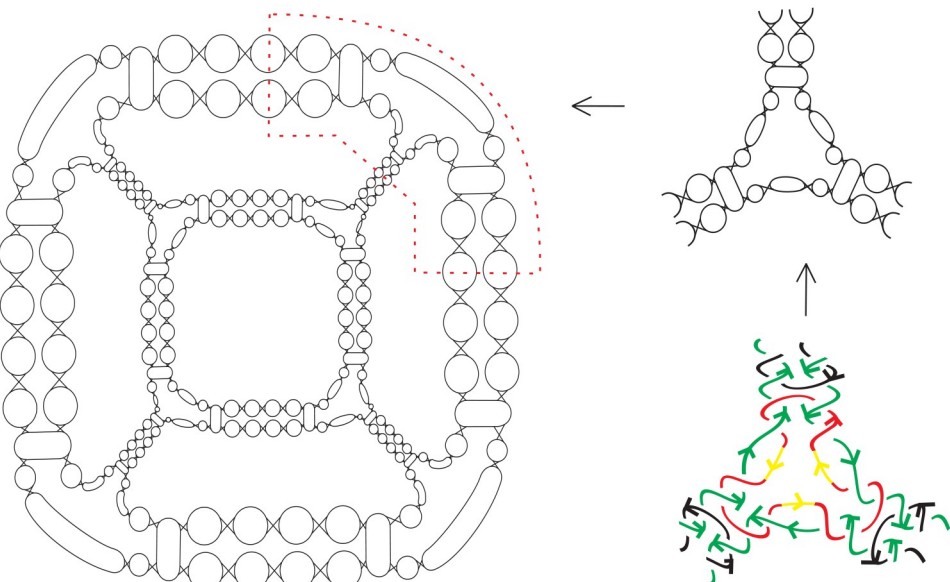

**Fig 9. The Seifert circle decomposition of the link $\mathcal{L}_0^*$ (as shown in Fig 3) whose Seifert graph corresponds to $G^*$ on the top of Fig 8**: each simple closed curve in the figure (drawn in different sizes and shapes, with the crossings ignored) is a Seifert circle.

$L(G)$ is a Type B double crossover polyhedral link with G as its template graph and by replacing its edges by Type (1B) cycles of length $2m_1$ with $m_1 \geq 2$, and replacing its vertices by Type (2) cycles of length $2km_2$ with $m_2 \geq 1$, then $b(L(G)) = (km_2 + 1)n(G) + (m_1 - 1)e(G) + f(G) - 1$, where $f(G)$ is the number of faces of G.

We shall delay the proof of Theorem 1 to Section 4. In the following we state a few results that are immediate consequences of Theorem 1. These include the previously open case of double crossover polyhedral links with 4.5 turn.

**Corollary 1** *Let G be a simple, k-regular plane bipartite graph. Let $L(G)$ be a double crossover polyhedral links with 4.5 turn, then $b(L(G)) = (2k + 1)n(G) + 4e(G) = \left(8 + \frac{2}{k}\right)e(G)$. On the other hand, if $L(G)$ is a double crossover polyhedral links with 4 turn, then $b(L(G)) = (2k + 1)n(G) + 3e(G) + f(G) - 1 = 2kn(G) + 4e(G) + 1 = 8e(G) + 1$.*

Notice that in the case of 4.5 turn and $k = 3$, $c(L(G)) = e(G^*) = 12n(G) + 10e(G) = 18e(G)$ hence $b(L(G)) = (13/27)c(L(G))$. In the case of 4 turn, $c(L(G)) = 12n(G) + 8e(G) = 16e(G)$ hence $b(L(G)) = (1/2)c(L(G)) + 1$.

**Proof**. In the case of 4.5 turn, the Type (1A) cycles are used and have length 10. The Type (2) cycles have length 12. Thus $m_1 = 5$ and $m_2 = 2$, it follows that

$$\begin{aligned} \mathbf{b}(L(G)) &= (km_2 + 1)n(G) + (m_1 - 1)e(G) \\ &= (2k + 1)n(G) + 4e(G). \end{aligned} \tag{6}$$

In the case of 4 turn, Type (1B) cycles are used and have length 8. Thus $m_1 = 4$ and $m_2 = 2$, it follows that

$$\begin{aligned} \mathbf{b}(L(G)) &= (km_2 + 1)n(G) + (m_1 - 1)e(G) + f(G) - 1 \\ &= (2k + 1)n(G) + 3e(G) + f(G) - 1. \end{aligned} \tag{7}$$

By the fact that $n(G) = (2e(G))/k$ for any k-regular graph G and the Euler's formula $n(G) + f(G) = e(G) + 2$, we can then simplify (6) to

$$\mathbf{b}(L(G)) = (2k + 1)n(G) + 4e(G) = \left(8 + \frac{2}{k}\right)e(G)$$

and (7) to

$$\begin{aligned} \mathbf{b}(L(G)) &= (2k + 1)n(G) + 3e(G) + f(G) - 1 \\ &= (2k + 1)n(G) + 3e(G) + e(G) - n(G) + 1 \\ &= 8e(G) + 1. \end{aligned}$$

For example, for the link $\mathcal{L}_0^*$ given in Fig 3, we have $e(G) = 12$, thus $b(L(G)) = \left(8 + \frac{2}{3}\right) \times 12 = 104$. On the other hand, if $L(G)$ is the link $\mathcal{L}_0$ with 4 turn given in Fig 2, then we obtain $b(L(G)) = 8 \times 12 + 1 = 97$, which is the same as $(1/2)c(L(G)) + 1$ since $c(L(G)) = 192$.

## 4 The main results and proofs for the general cases

Let $L(G)$ be a Type A or Type B double crossover polyhedral link constructed from the template graph G. Let us start this section by first introducing an operation called a *reduction move*. The total sum of all reduction move is called by *reduction number*. The top of Fig 10 illustrates how a strand can be re-routed to obtain a new diagram with one less Seifert circle. The middle of Fig 10 shows the effect of a reduction move on the corresponding Seifert graph when the middle vertex is not an attaching vertex. Notice that the reduction move affects three

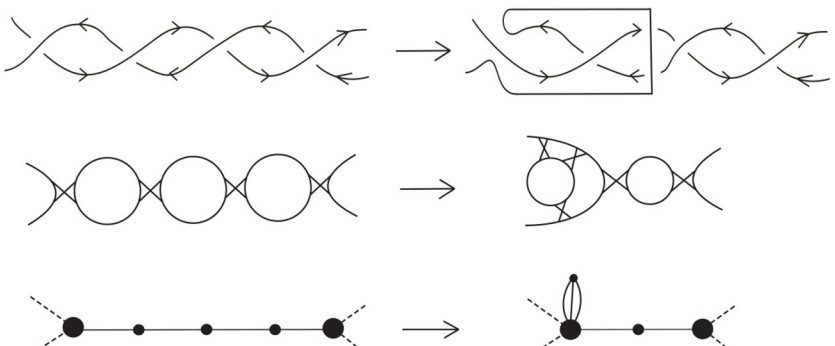

**Fig 10. The re-routing of the top strand at a single crossing reduces the number of Seifert circles by one.**

vertices on the Seifert graph of the link and the middle vertex cannot be an attaching vertex. The total number of such reduction moves we can take on $G^*$ is called the *reduction number* of $L(G)$ and denoted by $r(L(G))$. In the following we would like to determine $r(L(G))$.

In the case that $L(G)$ is a Type A double crossover polyhedral link, the reduction operation can be repeated $j$ times on a path of length $2j + 1$ connecting two attaching vertices. The top of Fig 11 shows an example of how a path of length 5 connecting two (distinct) attaching vertices is reduced to a single edge connecting the two attaching vertices, with two vertices attached (by multiple edges) to one of the attaching vertices. Thus in the case of a Type (1A) cycle $C_j$ with $2j_1 + 1$ and $2j_2 + 1$ edges on the two paths connecting the two attaching vertices, we can perform exactly $j_1 + j_2 = \ell(C_j)/2 - 1$ reduction moves (where $\ell(C_j) = 2j_1 + 2j_2 + 2$ is the length of $C_j$), and in the Seifert graph of the resulting diagram, the two affected attaching vertices are connected by two edges hence no more reduction moves can be made on $C_j$. So the reduction number of $C_j$ is $r(C_j) = j_1 + j_2 = \frac{\ell(C_j)}{2} - 1$, which is the contribution of $C_j$ to $r(L(G))$.

On the other hand, in the case of a path of even length $2j \geq 2$ connecting two distinct attaching vertices, $j$ reduction moves can be applied and the result is a "petal graph" with the leaf vertices attached to the attaching vertex as shown in the middle of Fig 11. Finally, in

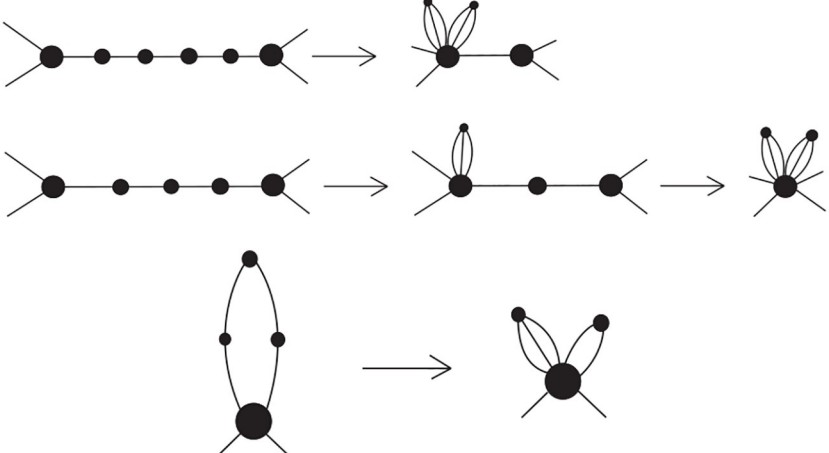

**Fig 11. How the repeated reduction move changes a path of odd length (top) and even length (middle), and a cycle (bottom).**

the case of a cycle of length $2j \geq 2$ with only one attaching vertex, $j - 1$ reduction moves can be applied and the result is also a "petal graph" with the leaf vertices attached to the attaching vertex as shown in the bottom of Fig 11. Thus for a Type (2) cycle $C_j$ with $2j_1$, $2j_2, \ldots, 2j_{k_j}$ edges on the paths connecting the adjacent attaching vertices, we can apply $r(C_j) = -1 + \sum_{1 \leq i \leq k_j} j_i = \frac{\ell(C_j)}{2} - 1$ reduction moves and the resulting Seifert graph is a petal graph.

It follows that if $L(G)$ is a Type A double crossover polyhedral link with $C_1, C_2, \ldots, C_k$ being the Type (1A) and Type (2) cycles used in its construction ($k = n(G) + e(G)$), then

$$r(L(G)) = -k + \sum_{1 \leq j \leq k} \frac{\ell(C_j)}{2} \tag{8}$$

with $\ell(C_1), \ell(C_2), \ldots, \ell(C_k)$ being the lengths of $C_1, C_2, \ldots, C_k$.

In the case that $L(G)$ is a Type B double crossover polyhedral link with $C'_1, C'_2, \ldots, C'_{k'}$ ($k' = e(G)$) being the Type (1B) cycles and $C''_1, C''_2, \ldots, C''_{k''}$ ($k'' = n(G)$) being the Type (2) cycles used in its construction, then each $C''_j$ still contributes $\ell(C''_j)/2 - 1$ to $r(L(G))$ as before. If we choose any spanning tree $T$ of $G$, then for each Type (1B) cycle $C'_j$ used to replace an edge on $T$, we can perform $\ell(C'_j)/2 - 1$ reduction moves. At the end all attaching vertices in $G^*$ are combined into a single attaching vertex. For any Type (1B) cycle $C'_j$ that is used to replace an edge of $G$ that is not on $T$, the two paths connecting its two attaching vertices now have both become cycles with one attaching vertex, hence we can only perform a total of $\ell(C'_j)/2 - 2$ reduction moves. Since $G$ has $f(G) - 1$ edges not on $T$, the total contribution of the Type (1B) cycles to $r(L(G))$ is $-f(G) + 1 + \sum_{1 \leq j \leq k'} r(C'_j)$ (keep in mind that $r(C'_j) = \ell(C'_j)/2 - 1$). Notice that at the end we obtain a petal graph with its leaf vertices attached to a single attaching vertex by multiple edges. Thus if we rename the Type (1B) and Type (2) cycles as $C_1, C_2, \ldots, C_k$ ($k = k' + k'' = e(G) + n(G)$), then

$$r(L(G)) = -f(G) + 1 + \sum_{1 \leq j \leq k} \left( \frac{\ell(C_j)}{2} - 1 \right), \tag{9}$$

where $f(G)$ is the number of faces in $G$.

The above discussion then leads to the following lemma.

**Lemma 1** *Let $L(G)$ and $r(L(G))$ be as defined in (8) or (9), then $L(G)$ admits a different link diagram $L'$ such that $s(L') = s(L(G)) - r(L(G))$, or equivalently, $n(G') = n(G) - r(L(G))$, where $G'$ is the Seifert graph of $L'$.*

Now let us consider a sequence of three consecutive Seifert circles connected by single crossings as shown in Fig 12 as part of an alternating link diagram. If one crossing is "flipped", then a Reidemeister move II can be applied afterward to reduce the number of crossings by 2. This reduces the number of Seifert circles by 2 and the resulting link diagram is still alternating, whose corresponding Seifert graph is obtained from the previous one by contracting the

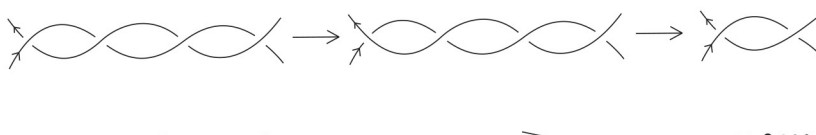

**Fig 12. Flipping a crossing followed by a Reidemeister move II reduces the number of Seifert circles by two, and corresponding to a special contraction on the corresponding Seifert graph.**

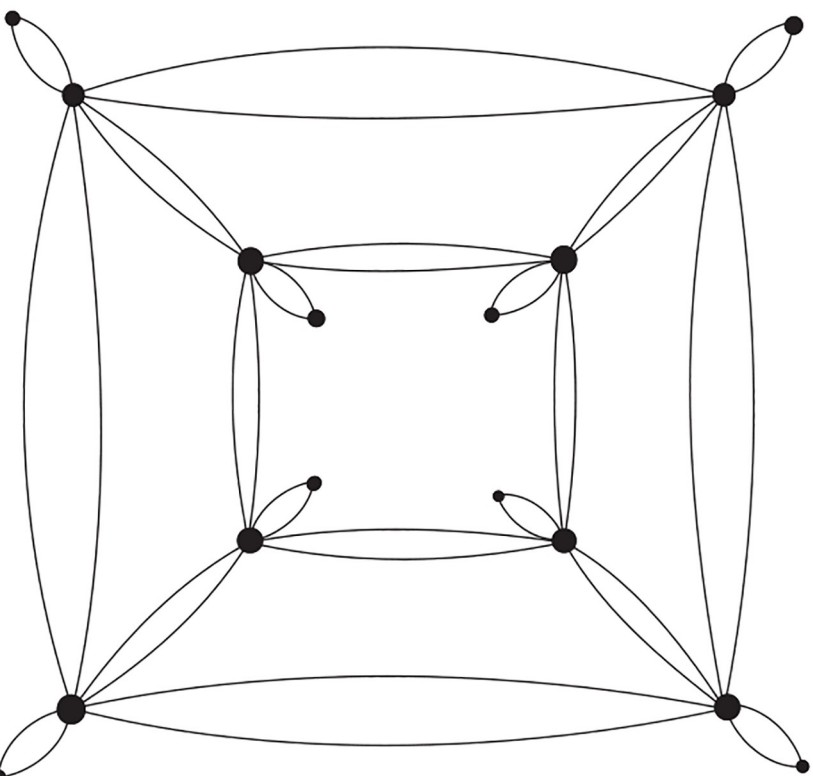

**Fig 13. The end product of the Seifert graph of the link $\mathcal{L}_0^*$ given in Fig 3, after all possible special reductions are performed.**

two edges as shown in Fig 12. Let us call the above operation on the link diagram this a *special contraction* if the middle vertex is not an attaching vertex.

By comparing the special contraction with the reduction move (defined in the proof of Lemma 1), we have the following two cases.

(a) If $L(G)$ is a Type A double crossover polyhedral link, then we can perform exactly $r(L(G))$ special contractions, and the resulting link diagram contains no *lone crossings* (a lone crossing is the only crossing between two Seifert circles). The resulting Seifert graph $G_0$ is the graph obtained from $G$ by doubling each edge of $G$ into two parallel edges, and attaching a vertex to each vertex of $G$ by two edges. An example is shown in Fig 13 for the link $\mathcal{L}_0^*$ given in Fig 3.

(b) If $L(G)$ is a Type B double crossover polyhedral link, we can also perform $r(L(G))$ special contractions where $r(L(G))$ is defined in (9). The Seifert graph of the resulting link diagram is a petal graph with its leaf vertices attached to a single attaching vertex by multiple edges. Fig 14 shows an example of a template graph $G$, a link diagram constructed from it using Type (1B) and Type (2) cycles, and how its Seifert graph is changed to a petal graph by performing $r(L(G))$ special contractions.

Let $q = r(L(G))$, $D^{(q)} = L(G)$, $D^{(q-1)}$ be the (alternating) link diagram obtained from $D^{(q)}$ by performing one special contraction, $D^{(q-2)}$ be the link diagram obtained from $D^{(q-1)}$ by performing one special contraction, and so on, and finally $D^{(0)}$ be the link diagram obtained from $D^{(1)}$ by performing the last available special contraction. We shall call the link diagrams $D^j$ ($0 \leq j \leq q$) *pseudo double crossover polyhedral links*. It is clear that the reduction number of $D^{(p)}$ ($0 \leq p \leq q$) is $p$. Furthermore, $L(G)$ and any $D^{(p)}$ are the *special* alternating link diagram

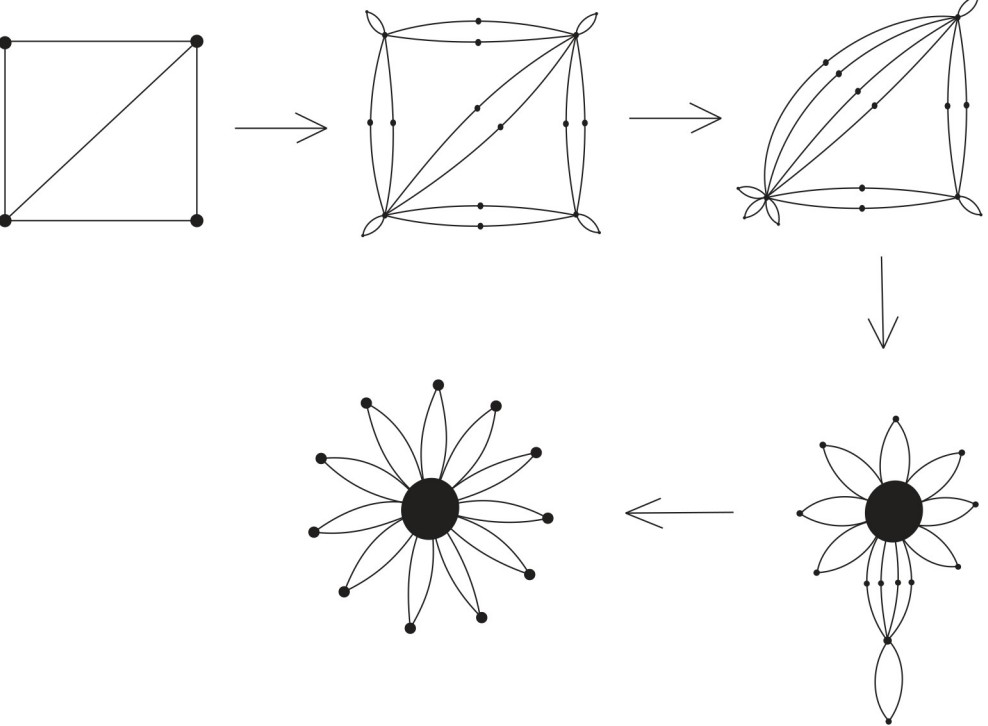

**Fig 14. The process of reducing the Seifert graph $G^*$ of a Type B double crossover polyhedral link $L(G)$ to a petal graph using special contractions.**

as defined in [31]. As such, all crossings in $L(G)$ and $D^{(p)}$ have the same sign. Since passing to the mirror image of a link does not change the braid index, we can assume all crossings in $L(G)$ are positive in the rest of this section. Thus the main theorem (Theorem 1) holds as well when the crossings in $L(G)$ are negative. Since the following well-known lemma is needed in the proof of our main result (Theorem 1), we state it here for completeness.

**Lemma 2** [32, 33] *Let L be any link diagram with only positive crossings, then $\gamma(L) = s(L) - w(L) - 1$, where $w(L)$ is the writhe of L.*

Recall that the writhe $w(L)$ of $L$ is simply the sum of the signs ($\pm 1$) of all crossings in $L$. Since $L$ has only positive crossings, we have $w(L) = c(L)$ where $c(L)$ is the number of crossings in $L$.

**Lemma 3** *Let L be any link diagram with only positive crossings and let $r(L)$ be the maximum number of Seifert circles that can be reduced by the reduction operations as described in* Fig 10, *then $\alpha(L) \geq -s(L) - w(L) + 1 + 2r(L)$. Furthermore, if $\alpha(L) = -s(L) - w(L) + 1 + 2r(L)$, then $\boldsymbol{b}(L) = s(L) - r(L)$.*

**Proof**. By Yamada [24] and Lemma 1, we have $\boldsymbol{b}(L) \leq s(L) - r(L)$. Combine this with Lemma 2 and the MFW inequality, we have

$$\gamma(L) - \alpha(L) + 2 \leq 2\boldsymbol{b}(L) \leq 2s(L) - 2r(L).$$

It follows that

$$\begin{aligned} \alpha(L) &\geq \gamma(L) + 2 - 2s(L) + 2r(L) \\ &= s(L) - w(L) - 1 + 2 - 2s(L) + 2r(L) \\ &= -s(L) - w(L) + 1 + 2r(L). \end{aligned}$$

If $\alpha(L) = -s(L) - w(L) + 1 + 2r(L)$, then the MFW inequality becomes $\mathbf{b}(L) \geq s(L) - r(L)$, the last statement of the lemma follows since we also have $\mathbf{b}(L) \leq s(L) - r(L)$.

Finally, before we state and prove our main result of this section (namely Theorem 2), we state the following lemma. It is a known result. A proof of this for general link diagrams can be found in [32], while a proof for the case of special alternating link diagrams can be found in [31].

**Lemma 4** [32, 33] *If L is an alternating link diagram such that its Seifert graph does not contain any single edge, that is, two vertices in its Seifert graph either are not connected by any edge, or are connected by more than one edge, then $\gamma(L) = s(L) - w(L) - 1$ and $\alpha(L) = -s(L) - w(L) + 1$. In particular, $\mathbf{b}(L) = s(L)$.*

**Theorem 2** *Let D be a pseudo double crossover polyhedral link, then $\mathbf{b}(D) = s(D) - r(D)$, where $r(D)$ is defined by either* (8) *or* (9), *depending on the type of the double crossover polyhedral link.*

**Proof**. Let $q = r(L(G))$, $D^{(q)} = L(G)$, $D^{(q-1)}, D^{(q-2)}, \ldots, D^{(1)}, D^{(0)}$ be the pseudo double crossover polyhedral links obtained from $L(G)$. Recall that the reduction number of $D^{(p)}$ ($0 \leq p \leq q$) is $p$. It suffices to prove that $\alpha(D^{(p)}) = -s(D^{(p)}) - w(D^{(p)}) + 1 + 2p$ for any $p$, $0 \leq p \leq q$. We will prove this by induction on $p$.

If $p = 0$, $D^{(0)}$ is a positive alternating link diagram without any lone crossings, and the statement follows from Lemma 4.

Assume that the induction hypothesis holds for some $p_0 \geq 0$, that is, $\alpha(D^{(p)}) = -s(D^{(p)}) - w(D^{(p)}) + 1 + 2p$ holds for any $p$ such that $0 \leq p \leq p_0$, to prove the theorem, it suffices for us to show that $\alpha(D^{(p_0 + 1)}) = -s(D^{(p_0 + 1)}) - w(D^{(p_0 + 1)}) + 1 + 2(p_0 + 1)$.

Choose a crossing corresponding to an edge in the special contraction taking $D = D^{(p_0 + 1)}$ to $D^{(p_0)}$ and apply the skein relation (4) to it. Notice that $D = D_+ = D^{(p_0 + 1)}$ and $D_-$ simplifies (via a Reidemeister II move) to $D^{(p_0)}$ with $s(\tilde{D}_-) = s(D) - 2$ and $w(\tilde{D}_-) = w(D) - 2$. As an easy exercise, we leave it to our reader to verify that $r(D_0) \geq p_0 + 2$, $s(D_0) = s(D)$ and $w(D_0) = w(D) - 1$. By the induction hypothesis we have:

$$
\begin{aligned}
-2 + \alpha(D_-) &= -2 + (-s(\tilde{D}_-) - w(\tilde{D}_-) + 1 + 2r(\tilde{D}_-)) \\
&= -2 - (s(D) - 2) - (w(D) - 2) + 1 + 2p_0 \\
&= -s(D) - w(D) + 1 + 2(p_0 + 1).
\end{aligned}
$$

On the other hand, by Lemma 3 we have:

$$
\begin{aligned}
-1 + \alpha(D_0) &\geq -1 + (-s(D_0) - w(D_0) + 1 + 2r(D_0)) \\
&\geq -1 - s(D) - (w(D) - 1) + 1 + 2(p_0 + 2) \\
&= -s(D) - w(D) + 3 + 2(p_0 + 1) \\
&> -s(D) - w(D) + 1 + 2(p_0 + 1).
\end{aligned}
$$

It follows that $a^{-2}P_{D_-}(a, z)$ is the only term on the right side of (4) making the lowest power $\alpha(D) = -s(D) - w(D) + 1 + 2(p_0 + 1)$ contribution to $P_D(a, z)$, hence $\alpha(D) = -s(D) - w(D) + 1 + 2(p_0 + 1)$, that is, $\alpha(D^{(p_0 + 1)}) = -s(D^{(p_0 + 1)}) - w(D^{(p_0 + 1)}) + 1 + 2(p_0 + 1)$.

Theorem 2 enables us to compute the braid index of any pseudo polyhedral link easily since the reduction number of such a link is easy to find. Furthermore, since the set of all double crossover polyhedral links with any given turn number is a subset of the set of all pseudo polyhedral links, the proof of Theorem 1 is a simple application of it, which we give below. In particular, for specific double crossover polyhedral links constructed using either special template graphs such as regular graphs, or double crossover polyhedral links with specific turn

numbers, their braid indices can be formulated using information only depending on the template graphs.

*Proof* of Theorem 1. In the case that Type (1A) cycles are used, we have $s(L(G)) = (2m_1)e(G) + 2km_2n(G) - 2e(G) = 2(m_1 - 1)e(G) + 2km_2n(G)$. By Lemma 1, $r(L(G)) = (m_1 - 1)e(G) + (km_2 - 1)n(G)$. It follows that

$$
\begin{aligned}
\mathbf{b}(L(G)) &= s(L(G)) - r(L(G)) \\
&= 2(m_1 - 1)e(G) + 2km_2n(G) \\
&\quad -(m_1 - 1)e(G) - (km_2 - 1)n(G) \\
&= (km_2 + 1)n(G) + m_1 e(G).
\end{aligned}
$$

On the other hand, if Type (1B) cycles are used, then by the definition (9) of $r(L(G))$, $r(L(G)) = (m_1 - 1)e(G) + (km_2 - 1)n(G) - f(G)$. Again by Lemma 1 and Theorem 2, we have

$$
\begin{aligned}
\mathbf{b}(L(G)) &= s(L(G)) - r(L(G)) \\
&= (km_2 + 1)n(G) + m_1 e(G) + f(G),
\end{aligned}
$$

as desired.

## 5 Further discussions and ending remarks

In this paper, we provide explicit braid index formula for a large class of links that include all double crossover polyhedral links, an application of this result leads to the solution of a braid index problem unreachable by previous approaches such as the method used in [21]. As we had already mentioned, Theorem 1 is valid for any double crossover polyhedral link with any given number turn. That is, the braid index of a double crossover polyhedral link with any given number turn has now been completely determined. We would like to compare the results obtained here with some previous results.

In [21], Cheng and Jin studied a class of double crossover polyhedral links with 4 turn based on connected, bridgeless and loopless plane template graph $G$ (the DNA polyhedra corresponding to these links have been synthesized [10, 16–20]). A main result in [21] is the following theorem which relates the braid index of $L(G)$ to its minimum crossing number $c(L(G))$ by a simple formula.

**Theorem 3** [21] *Let $G$ be a connected, bridgeless and loopless plane graph and $L(G)$ a double crossover polyhedral links with 4 turn using $G$ as its template graph, then $\mathbf{b}(L(G)) = c(L(G))/2 + 1$.*

A special case of this result has already been established in Corollary 1. We will provide a proof of this more general result using Theorem 2.

**Proof**. Let $v_1, v_2, \ldots, v_n$ be the vertices of $G$ with degrees $d_1, d_2, \ldots, d_n$ respectively (where $n = n(G)$). Keep in mind for a double crossover polyhedral links with 4 turn, Type (1B) cycles of length 8 are used to replace edges of $G$, and the path between any two adjacent attaching vertices of a Type (2) cycle is of length 4. Thus each edge of $G$ contributes 8 crossings to $c(L(G))$ and a vertex of degree $d_j$ contribute $4d_j$ crossings to $c(L(G))$. So we have

$$
c(L(G)) = (4d_1 + 4d_2 + \cdots + 4d_n) + 8e(G) = 16e(G),
$$

where $e(G)$ is the number of edges in $G$. Similarly, we have

$$
s(L(G)) = (4d_1 + 4d_2 + \cdots + 4d_n) + 6e(G) = 14e(G).
$$

On the other hand, each edge of $G$ corresponds to a Type (1B) cycle of length 8, hence it makes a contribution of 3 to the reduction number of $L(G)$, while a vertex of degree $d_j$ corresponds to

a Type (2) cycle of length $4d_j$, hence making a contribution of $2d_j - 1$ to the reduction number of $L(G)$. Thus (by the definition of $r(L(G))$, since Type (1B) cycles are used):

$$\begin{aligned} r(L(G)) &= 3e(G) + \sum_{1 \leq j \leq n} (2d_j - 1) - f(G) + 1 \\ &= 7e(G) - n(G) - f(G) + 1. \end{aligned}$$

It follows that

$$\begin{aligned} \mathbf{b}(L(G)) &= s(L(G)) - r(L(G)) \\ &= 14e(G) - 7e(G) + n(G) + f(G) - 1 \\ &= 7e(G) + n(G) + f(G) - 1. \end{aligned}$$

By Euler's formula, $n(G) - e(G) + f(G) = 2$, hence $n(G) + f(G) = e(G) + 2$ and we arrive at

$$\begin{aligned} \mathbf{b}(L(G)) &= 7e(G) + n(G) + f(G) - 1 \\ &= 7e(G) + e(G) + 1 \\ &= 8e(G) + 1 \\ &= \frac{c(L(G))}{2} + 1. \end{aligned}$$

In the case that Type (1A) cycles are used,

$$c(L(G)) = (4d_1 + 4d_2 + \cdots + 4d_n) + 10e(G) = 18e(G),$$

$$s(L(G)) = (4d_1 + 4d_2 + \cdots + 4d_n) + 8e(G) = 16e(G)$$

and

$$\begin{aligned} r(L(G)) &= 4e(G) + \sum_{1 \leq j \leq n} (2d_j - 1) \\ &= 8e(G) - n(G). \end{aligned}$$

A proof similar to the proof of Theorem 3 leads to the following new result, which is summarized in Theorem 4.

$$\begin{aligned} \mathbf{b}(L(G)) &= s(L(G)) - r(L(G)) \\ &= 16e(G) - 8e(G) + n(G) \\ &= 8e(G) + n(G) \\ &= 9e(G) - f(G) + 2 \\ &= \frac{c(L(G))}{2} - f(G) + 2. \end{aligned}$$

**Theorem 4** *Let G be a connected, bipartite plane graph and L(G) a double crossover polyhedral links with 4.5 turn using G as its template graph, then* $\mathbf{b}(L(G)) = c(L(G))/2 - f(G) + 2$, *where c(L(G)) is the crossing number of L(G) and f(G) is the number of faces of G.*

If we apply this theorem to $\mathcal{L}_0^*$ (given in Fig 3), then $c(L(G)) = 18e(G) = 18 \times 12 = 216$, $f(G) = 6$ hence $\mathbf{b}(L(G)) = c(L(G))/2 - f(G) + 2 = 108 - 6 + 2 = 104$, as we have obtained earlier after Corollary 1.

We end this paper with the following remark. As we mentioned earlier, the class of pseudo polyhedral links is large compared to the double crossover ones. For example, for each

template graph $G$ that is $k$-regular, there is exactly one double crossover polyhedral link $L(G)$ with 4.5 turn using $G$ as the template graph (with $c(L(G)) = 4kn(G) + 10e(G) = 18e(G)$), but the number of pseudo Type A double crossover polyhedral links with $G$ as their template graph and with $18e(G)$ as their crossing number can be roughly estimated in the order of $3^{3e(G)}$. A precise enumeration is beyond the scope of this paper and will be addressed in a future work. This means that in general, the number of pseudo double crossover polyhedral links grows exponentially as a function of the crossing number. However, we need to point out that the class of pseudo polyhedral links is nonetheless a very special class of links among all links of the same crossing number. It is a subset of the intersection of several well known classes of links: the alternating links, the special links (defined and studied by Murasugi in [30]) and the positive (negative) links. We note that although there have been studies concerning the braid index of links in these classes, these studies do not contain results that can be readily applied to the pseudo double crossover polyhedral links.

## Acknowledgments

The first author wishes to thank Professor Xian'an Jin for his support, encouragement and helpful discussions in the past that led the author to this current research.

## Author Contributions

**Conceptualization:** Xiao-Sheng Cheng, Yuanan Diao.

**Formal analysis:** Xiao-Sheng Cheng.

**Methodology:** Xiao-Sheng Cheng, Yuanan Diao.

**Writing – original draft:** Xiao-Sheng Cheng, Yuanan Diao.

**Writing – review & editing:** Xiao-Sheng Cheng, Yuanan Diao.

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
