## [Decision Letter · Decision Letter 0]

14 Jan 2020

PONE-D-19-31218

The braid index of DNA double crossover polyhedral links

PLOS ONE

Dear Dr Diao,

Thank you for submitting your manuscript to PLOS ONE. After careful consideration, we feel that it has merit but does not fully meet PLOS ONE’s publication criteria as it currently stands. Therefore, we invite you to submit a revised version of the manuscript that addresses the points raised during the review process.

We would appreciate receiving your revised manuscript by Feb 28 2020 11:59PM. To enhance the reproducibility of your results, we recommend that if applicable you deposit your laboratory protocols in protocols.io, where a protocol can be assigned its own identifier (DOI) such that it can be cited independently in the future. For instructions see: http://journals.plos.org/plosone/s/submission-guidelines#loc-laboratory-protocols

We look forward to receiving your revised manuscript.

Kind regards,

Akbar Ali

Academic Editor

PLOS ONE

Journal Requirements:

Reviewers' comments:

Reviewer's Responses to Questions

**Comments to the Author**

1. Is the manuscript technically sound, and do the data support the conclusions?

Reviewer #1: Yes

Reviewer #2: Yes

2. Has the statistical analysis been performed appropriately and rigorously? 

Reviewer #1: N/A

Reviewer #2: N/A

3. Have the authors made all data underlying the findings in their manuscript fully available?

Reviewer #1: Yes

Reviewer #2: Yes

4. Is the manuscript presented in an intelligible fashion and written in standard English?

Reviewer #1: Yes

Reviewer #2: Yes

5. Review Comments to the Author

Reviewer #1: In the past 30 years, many DNA or protein polyhedra have been realized in the lab. Anaylizing their strucute and further measuring them leaves a difficut task for most biochemists. In this paper, the authors used the braid index in knot theory to measure the complexity of a family of DNA polyhedra with 4.5 turns. The tools they used are feasible and the results are new and sound correct.

The paper no dout deserves publication, but the authors should check the details in mathematical proof carefully again.

Reviewer #2: This paper provides theoretical foundations for computing the braid index of a certain class of knots that play a role in synthesizing DNA molecules in laboratories as outlined in the introduction. The theoretical importance of the braid index stems from Alexander's theorem saying that every knot can be represented as a closure of some braid. From experiments it appears that the braid form of a knot provides a good way to build complicated knots by using certain braids as building blocks. The starting point then is to know what is the minimal number of strands of the braid, known as the braid index, needed to represent a particular knot in the braid form. The authors hint at the possibility that the braid index can be used as a measure of the complexity of the molecule.

The main result of the paper states that for a certain class of knots that contains the double crossover polyhedral links, the famous Morton-Franks-Williams inequality, relating the span of the HOMFLY-PT polynomial with the braid index, is actually an equality. The proof, provided in Section 4, comprises of a clever use of ideas from combinatorial topology, relating topological properties of knots such as the braid index and the number of Seifert circles, to properties of certain graphs associated to knots.

This result answers and open question in knot theory and provides theory behind experiments that have already been successfully carried out, such as generating a double crossover polyhedral link with a 4.5 turn but the braid index was not known. The main result of this paper computes the braid index of double crossover polyhedral link with any turn number, and also the class of all pseudo polyhedral links. It also provides a useful framework for reproving and generalizing results relating the braid index with the crossing number of a knot (Theorems 3 and 4).

General suggestions:

- It would be great if the authors could provide the description and comparisons of classes of knots that are at play, e.g. how large is the class of pseudo polyhedral knots compered to the double crossover ones? What are other topological properties of these knots that are known or of interest for applications?

- Can Section 3 have a more specific title or would the authors consider adding something to the title that would say that the results apply to a larger class of knots and links?

- Titles of Sections 3 and 4 seem indistinguishable. Is it fair to specify that Section 4 contains the proof?

- Suggestion: include notation for the link on Figure 3 and use it as a running example as it appears several times in the text, including page 9 (where in the paragraph starting with "For example, for the double.." it is not at all clear which graph G is used so it would be good to be able to refer to it).

Specific comments:

- Pg 2. Replace "terminologies" with terminology

- Pg 3. and Pg 5. Replace "It is well knowns" with something stating that according to Alexander's theorem

- Pg 3. Up to the authors but we suggest using HOMFLY-PT instead of HOMFLY to acknowledge all of the independent constructions

- Pg. 7 Fig. 7 Please add color or labels describing the correpsondence between Seifert circles and the vertices of the graph, crossings and edges... also add notation from the section 2.3 to the caption of Figure 7.

-Pg 7. "We know" can be safely removed from the first sentence of Ch 3.

-Pg 8. It is impossible to distinguish different Seifert circles in Figure 9. Color or some other tool should be used to make this clear.

-Pg 8 Caption of figure 9: please specify which link is in Figure 3 and change "in the top" to "on the top"

-Pg 9. "For example, for the double.." see general suggestions but definitely include figures.

-Pg 9. "Similarly " comes after examples- please specify

-Pg 9. "It is fairly easy" is not very useful as this usually implies a significant level of technicalities that the authors did not want to go into. Please modify or include a more specific description of what the generalized result would look like/depend on.

-Pg 10. In addition to changing the title please add a little description of this section in the beginning.

-Pg 11. Increase the size/thickness on Fig. 11, 13, and 14.

- pg 13: replace "the following lemma is well known .." with "Since the following well-known lemma is needed in the proof of the main Theorem 1 we state it here for completeness" or with a similar sentence.

- Pg 14. Theorem 2 provides a powerful tool for ? please make it a more complete sentence if possible. I am not sure how to parse "on the one hand" and "on the other hand" as these two statements read as "Theorem 2 is very powerful (at least partially) because the class of links it covers is large" and then it also has the main Theorem 1 as a simple corollary.

-Pg. 14. Can you provide more examples of knots and links in this class? Is there a database?

-Pg 15. stylistic suggestion: replace the first sentence of Section 5 with "In this paper we provide explicit formula" to avoid repeating the same phrase.

- Pg 16. Maybe replace: Thus we obtain ... with "the insights from the proof of Theorem 3...."

- Pg 16. Define all notation used in Theorem 4.

6. PLOS authors have the option to publish the peer review history of their article (what does this mean?). If published, this will include your full peer review and any attached files.

Reviewer #1: No

Reviewer #2: No

---

## [Author Response · Author response to Decision Letter 0]

20 Jan 2020

A detailed response to the reviewers comments is attached in the uploaded file named "revision report". To summarize: we have addressed all issued raised by the referees.

---

## [Editor Report · Decision Letter 1]

27 Jan 2020

The braid index of DNA double crossover polyhedral links

PONE-D-19-31218R1

Dear Dr. Diao,

We are pleased to inform you that your manuscript has been judged scientifically suitable for publication and will be formally accepted for publication once it complies with all outstanding technical requirements.

With kind regards,

Akbar Ali

Academic Editor

PLOS ONE
---

## [Editor Report · Acceptance letter]

30 Jan 2020

PONE-D-19-31218R1 

The braid index of DNA double crossover polyhedral links 

Dear Dr. Diao:

I am pleased to inform you that your manuscript has been deemed suitable for publication in PLOS ONE. Congratulations! Your manuscript is now with our production department. 

With kind regards,

on behalf of

Dr. Akbar Ali 

Academic Editor

PLOS ONE